# Chicken Embryo Fibroblast Viability and Trans-Differentiation Potential for Cultured Meat Production Across Passages

**DOI:** 10.3390/cells13201734

**Published:** 2024-10-19

**Authors:** So-Hee Kim, Chan-Jin Kim, Eun-Yeong Lee, Young-Hwa Hwang, Seon-Tea Joo

**Affiliations:** 1Division of Applied Life Science (BK21 Four), Gyeongsang National University, Jinju 52828, Republic of Korea; winight123@gmail.com (S.-H.K.); ckswls09090@gmail.com (C.-J.K.); ley9604@gmail.com (E.-Y.L.); 2Institute of Agriculture & Life Science, Gyeongsang National University, Jinju 52828, Republic of Korea; philoria@gnu.ac.kr

**Keywords:** chicken embryo fibroblast, cultured fat, cultured meat, adipogenic trans-differentiation, adipocyte, primary fibroblast, passage

## Abstract

This study was conducted to analyze the viability of primary chicken embryo fibroblasts and the efficiency of adipogenic trans-differentiation for cultured meat production. In isolating chicken embryo fibroblasts (CEFs) from a heterogeneous cell pool containing chicken satellite cells (CSCs), over 90% of CEFs expressed CD29 and vimentin. The analysis of the proliferative capabilities of CEFs revealed no significant differences in EdU-positive cells (%), cumulative cell number, doubling time, and growth rate from passage 1 to passage 9 (*p* > 0.05). This indicates that CEFs can be isolated by 2 h of pre-plating and survive stably up to passage 9, and that primary fibroblasts can serve as a valuable cell source for the cultured meat industry. Adipogenic trans-differentiation was induced up to passage 9 of CEFs. As passages increased, lipid accumulation and adipocyte size significantly decreased (*p* < 0.05). The reduced differentiation rate of primary CEFs with increasing passages poses a major challenge to the cost and efficiency of cultured meat production. Thus, effective cell management and the maintenance of cellular characteristics for a long time are crucial for ensuring stable and efficient cultured fat production in the cultured meat industry.

## 1. Introduction

As fat is an important factor in determining the taste and profile of meat, adding fat is essential to manufacture cultured meat with characteristics similar to traditional meat [1]. To enhance the quality of cultured meat in terms of taste, flavor, and texture, researchers are undertaking numerous endeavors [2,3,4,5]. One study suggested that the taste of cultured tissue can vary depending on amino acid composition and umami substances and that the flavor of cultured meat can differ based on fat composition [6]. Therefore, the provision of fat is indispensable in the production of cultured meat. Various methods of adding fat have been applied to manufacture numerous types of cultured meat containing fat [7,8,9]. Among them, adding cultured fat has recently captured interest. Cultured fat can be supplied by culturing adipocytes, which can be obtained by isolating fat tissues or inducing trans-differentiation from other types of cells [10,11]. Trans-differentiation is a transition in which differentiated cells are irreversibly converted to another cell type without going through the progenitor cell [12]. It can be induced by changing the microenvironment of a cell or providing specific signaling molecules [13]. Several cell types originating from mesenchymal stem cells (MSCs), such as fibroblasts and osteoblasts, can be transformed into mature adipocytes through trans-differentiation [14,15]. However, it is necessary to adjust conditions or treatments, since transcription factors related to trans-differentiation are different depending on the animal species.

Fat can be efficiently added to cultured meat by trans-differentiation even when primary cells are used. To utilize primary cells for culture meat production, an essential process of sorting, which involves isolating targeted cells, is required. Ensuring the purity of cells destined to differentiate into muscle through cell sorting is also crucial for creating cultured tissues [16]. Several methods for sorting cells based on cellular characteristics have been studied [17,18,19,20]. Among them, pre-plating can isolate satellite cells and fibroblasts based on the difference in the adhesion time of cells [21]. Various cell species can be attached to the flask during the pre-plating step to increase the purity of satellite cells [22]. Pre-plating has an advantage in that cells that can rapidly attach to the flask can be trans-differentiated to adipocytes without any additional cell isolation. Most studies utilizing trans-differentiation to produce cultured fat have focused on the conversion of fibroblasts into adipocytes [10,23,24]. In addition, research has indicated that various cells such as myocytes and osteoblasts can trans-differentiate into adipocytes [10,14,25,26].

Fibroblasts are present throughout the animal body. They can be easily observed in the primary culture of satellite cells. They are suitable for adipogenic trans-differentiation during the production of cultured meat [27,28]. Fibroblasts can maintain their stemness for a long time in vitro. Human fibroblasts are known to be able to maintain potency for up to 30 passages [29]. Due to their high cell plasticity, fibroblasts are suitable for trans-differentiation, enabling them to transform into other cell types [13]. Based on these characteristics of fibroblasts, it is believed that cell yield will significantly increase, contributing to cultured meat production when fat is produced through trans-differentiation. Hence, this study aimed to explore for how many passages chicken embryonic fibroblasts (CEFs) could maintain their characteristics in vitro and to determine differences in the trans-differentiation of fibroblasts into adipocytes depending on the number of passages.

## 2. Materials and Methods

### 2.1. Animal Care and Research Ethics Statements

The treatment and use of experimental animals were approved by the Institutional Animal Care and Use Committee (IACUC) at Gyeongsang National University (approval no. GNU-231127-C0215). All experimental steps complied with the IACUC standard procedures.

### 2.2. Isolation of Chicken Embryo Fibroblasts (CEFs)

Hindlimb muscle tissues were isolated from 18-day-old chicken embryos, removing bones and cartilage as much as possible. These tissues were then washed and immersed in Hanks balanced salt solution (HBSS; Welgene, Gyeongsan, Republic of Korea) containing 1% antibiotic-antimycotic (anti-anti) (Gibco™, Carlsbad, CA, USA) to disinfect and eliminate extrinsic debris including hair and blood. The sample was minced with scissors until it became a thick liquid. The tissue sample was digested with 0.1% Type I collagenase; (Worthington Biochemical Corp. Lakewood, NJ, USA) and incubated at 37 °C for 1 h. During enzymatic digestion with collagenase, the suspension was aspirated and discharged from a syringe with an 18-gauge needle every 15 min. The digested tissue sample was centrifuged at 4 °C, 800× *g* for 5 min and the supernatant was removed. To obtain a single-cell suspension, 0.25% trypsin-EDTA (Gibco™, Carlsbad, CA, USA) was added to the cell pellet. The cell suspension with trypsin was incubated at 37 °C for 20 min. After digestion, fetal bovine serum (FBS) was added to the single-cell suspension to inhibit the action of trypsin. After Dulbecco’s phosphate-buffered saline (DPBS; Welgene, Gyeongsan, Republic of Korea) was added to the cell suspension, the cell suspension was mixed gently in a circular pattern. When a viscous lump started to rise, amorphous and flouting liquid was removed by aspirating with a syringe with an 18-gauge needle. During removal, the loss of cell suspension had to be minimized. The cell suspension was filtered using cell strainers with pore sizes in the order of 100 μm and 40 μm. It was then centrifuged at 4 °C at 800× *g* for 5 min. Red blood cell lysis buffer (Invitrogen^TM^, Waltham, MA, USA) was added to the pellet after removing the supernatant. The detailed experimental procedures followed the manufacturer’s guidelines.

### 2.3. Purification of Chicken Embryo Fibroblasts (CEFs)

The cell suspension isolated through primary culture was purified using the pre-plating method. Cells obtained from primary culture were incubated in uncoated flasks for 2 h. After this incubation, adherent cells were transferred to collagen-coated flasks and incubated for an additional 2 h. Following this incubation, adherent cells were identified as CEFs. The growth medium was subsequently replaced. The supernatant was transferred to another culture flask to start the main culture.

### 2.4. Culture Condition of Chicken Embryo Fibroblasts (CEFs)

Cells were seeded at 2500 cells/cm^2^ for both proliferation and differentiation. Growth media (GM; 20% FBS, 1% Glutamax supplement (Gibco™, Carlsbad, CA, USA) and 1% anti-anti in DMEM) were used for the proliferation of CEFs. The cells were incubated at 41 °C with 5% CO_2_ for proliferation. Myofibroblasts were cultured in trans-differentiation media (DM) supplemented with 10% FBS, 1% Glutamax supplement, 1% anti-anti in DMEM, 400 µM oleic acid (Santa Cruz Biotechnology, Dallas, TX, USA), and 5 µM or 10 µM rosiglitazone (Med Chem Express, Monmouth Junction, NJ, USA) to induce adipogenic trans-differentiation. GM was replaced with DM after 2 days of proliferation. The cells were incubated at 41 °C with 5% CO_2_. Adipogenic trans-differentiation was progressed for 14 days.

### 2.5. Immunofluorescence Staining for Isolating and Identifying Chicken Embryo Fibroblasts (CEFs)

Cells were fixed with 4% paraformaldehyde (PFA) for 15 min at room temperature (RT) and washed with DPBS for 5 min at least twice. Depending on the substance to be stained, permeabilization was conducted with 0.5% Triton X-100 for 15 min, followed by washing with DPBS for 5 min at least twice. After permeabilization, the blocking of cells was executed with 3% bovine serum albumin (BSA; Sigma Aldrich, St. Louis, MA, USA) in DPBS. The cells were incubated with primary antibodies including mouse anti-pax7 (Paired-box protein 7) conjugated with Alexa fluor 488 (1:200, Santa Cruz Biotechnology, Dallas, TX, USA) and mouse anti-MHC (1:40, DSHB, Iowa City, IA, USA) at 4 °C overnight. After washing with DPBS for 5 min twice after incubation, the cells were incubated with secondary antibodies, goat anti-mouse IgG2b cross-adsorbed secondary antibody conjugated with Alexa Fluor 488 (1:500, Thermo Scientific, Waltham, MA, USA) (diluted in 3% BSA in DPBS), for 2 h at RT in a dark room. Nuclei were counterstained with Hoechst 33342 (Invitrogen^TM^, Waltham, MA, USA). Immunofluorescence images were captured using an Olympus fluorescence microscope CKX 53 (EP50, Olympus, Hachioji-shi, Tokyo, Japan) and processed with ImageJ software (Version 1.54i, NIH, Bethesda, MD, USA). For statistical analysis, mean image data derived from five randomly selected images were used.

### 2.6. Proliferation Analysis of Chicken Embryo Fibroblasts (CEFs)

#### 2.6.1. Cumulative Cell Numbers

The cumulative cell number indicates cell growth rate by calculating the number of cells that have proliferated at each passage. The cumulative cell number of cells from passage 1 (P1) to passage 9 (P9) was calculated based on the following equation:(1)$$\sum_{n=1}\;\frac{N_{\left(n\right)}}{S_{\left(n\right)}}\times N_{\left(n-1\right)}$$
where *S* is the number of cells seeded, *N* is the final number of cells proliferated, and *N*_(0)_ is the initial number of cells.

#### 2.6.2. Population Doubling Time

Population doubling time (PDT) was calculated based on the following equation referenced from the method described previously [30]:(2)$$PDT=\frac{dT}{\log_{2}\left(\frac{N_{1}-N_{0}}{N_{0}}+1\right)}$$
where *N_0_* is the number of cells seeded, *N*_1_ is the final number of cells proliferated, and dT is the duration of culture. In the present study, the duration of culture (dT) was calculated by seeding 5000 cells/cm^2^ from passage 0 (P0) to passage 15 (P15).

#### 2.6.3. EdU Assay

An EdU (5-Ethynyl-2′-deoxyuridine) assay was conducted using Click-iT Plus EdU Imaging Kits (Invitrogen™, Waltham, MA, USA). Cells were incubated overnight after seeding. After incubation, 10 µM of EdU was added to live cells. Imaging was conducted after 4 h of EdU treatment. The detailed experimental procedures were conducted according to the manufacturer’s instructions. Mean image data derived from five randomly selected EdU assay images were presented for statistical analysis.

#### 2.6.4. Proliferation Capacity Assay Using Cell Counting Kit-8

The proliferation capacity of live cells was analyzed using Cell counting Kit-8 (CCK-8) (Dojindo, Kumamoto, Japan). The assay was conducted with eight replicates by culturing cells in a 96-well plate. After 24 h and 72 h of seeding cells, CCK-8 reagent was added to live cells. After incubating for 3 h, optical density (O.D) was measured at 450 nm with a microplate reader (Multiskan FC v1.0.74.0, Thermo Scientific, Waltham, MA, USA). The specific experimental procedures were performed according to the manufacturer’s instructions.

### 2.7. Adipogenic Trans-Differentiation Analysis over Passages

#### 2.7.1. Size Analysis of Transdifferentiated Adipocytes

Transdifferentiated chicken preadipocytes and adipocytes from passages 1, 5, and 9 were used to measure adipocyte size. The cells at each passage was cultured for 14 days. Adipocyte size analysis was conducted by selecting three randomly extracted microscope images, and the sizes of 20 cells from each passage were measured using ImageJ software.

#### 2.7.2. Neutral Lipid Staining

Neutral lipid in adipocytes was stained with HCS LipidTOX™ (Invitrogen™, Waltham, MA, USA). Cells were fixed with 4% formaldehyde solution in DPBS for 20 min at RT. After fixation, the cells were rinsed with DPBS two times to remove the remaining formaldehyde solution. LipidTOX diluted in DPBS at 1:500 was added to the cells. Staining was conducted at RT for 45 h. The detailed experimental procedures were performed according to the manufacturer’s instructions.

#### 2.7.3. Flow Cytometry for Adipogenesis Analysis

Trans-differentiated fibroblasts were stained with HCS LipidTOX and Hoechst 33342 according to the manufacturer’s instructions. The cells were rinsed with DPBS twice and resuspended in FACS buffer. Flow cytometry was performed using a FACS Analyzer (BD Biosciences, Franklin Lakes, NJ, USA).

### 2.8. Statistical Analysis

All statistical analyses were performed using SAS 9.4 (SAS, version 9.4, SAS Institute, Cary, NC, USA) and GraphPad Prism 10.3.2 (GraphPad version 10.2, GraphPad Software, San Diego, CA, USA). Statistical analysis was performed via one-way analysis of variance (ANOVA) with multiple comparisons. Tukey’s honestly significant difference (HSD) post hoc test was performed to compare means. All error bars in figures indicate SD. A *p*-value ≤ 0.05 indicated statistical significance.

## 3. Results

### 3.1. Isolation of Chicken Embryo Fibroblasts by Pre-Plating

A heterogeneous cell suspension obtained by filtering hindlimb muscle solution isolated from chicken embryos was enzymatically digested and sorted through pre-plating (Figure 1). Cells that adhered quickly to the flask within 2 h during pre-plating were classified as fibroblasts, while those sorted in subsequent pre-plating steps were categorized as CSCs. After cell purification, the attachment and morphological characteristics of CEFs were analyzed and compared to isolated CSCs. Both CEFs and CSCs were observed to have a similar morphology under the microscope on day 2 of culture (Figure 2A). However, when differentiation was induced for cells after starvation on day 6 of culture, CSCs formed myotubes whereas fibroblasts exhibited a distinct fibroblastic pattern, failing to form myotubes. This observation was consistent with the result of MHC expression, which was the lowest in fibroblasts (Figure 2B) (*p* < 0.05). Additionally, the doubling time of CEFs was 1.6 times shorter than that of CSCs (Figure 3C) (*p* < 0.05).

### 3.2. Identification of Chicken Embryo Fibroblasts

Cells were stained with several specific markers to identify fibroblasts isolated by pre-plating. Isolated CEFs and CSCs were immunofluorescence-stained with pax7 (Figure 3). CEFs showed 2% positivity for pax7, whereas 90% of cells purified through subsequent pre-plating steps expressed pax7 (Figure 3C) (*p* < 0.05). This is consistent with the results of previous morphological differences and doubling time variations among the cells (Figure 2). For additional identification, fibroblasts were stained with fibroblast markers CD29 and vimentin (Figure 3A). Stained images revealed that over 90% of fibroblasts expressed CD29 and vimentin (Figure 3C).

### 3.3. Analysis of Proliferation Capacity of Chicken Embryo Fibroblasts

The proliferative capacity of fibroblasts was evaluated through 15 passages to assess their suitability as materials for cultured meat production. Doubling times showed consistent and stable proliferation capacity across passages (Figure 4A). Except for cells at passage 0 immediately following primary culture, fibroblasts could maintain significant constant proliferation ability up to passage 12 (*p* < 0.05). The growth rate assessed by the CCK-8 assay showed no significant difference up to passage 9, indicating sustained proliferation and survival even at higher passages (Figure 4B) (*p* < 0.05). Moreover, cumulative cell numbers calculated based on the proliferated CEFs continuously increased up to passage 15 stably (Figure 4C). Additionally, cells at passages 1, 7, and 11 were stained with EdU to observe actively proliferating cells (Figure 4D). The proportion of cells expressing EdU did not show significant differences among passages (Figure 4D). However, from passage 11, doubling time and the CCK-8 assay revealed that the proliferation ability of fibroblasts was significantly decreased (*p* < 0.05).

### 3.4. Adipogenic Trans-Differentiation of Chicken Embryo Fibroblasts

Each passage of CEFs was induced for adipogenic trans-differentiation for a total of 14 days in DM (Figure 5A). For the initial 3 days post-differentiation, a DM containing 400 µM of oleic acid was utilized, followed by differentiation induction with additional 5 µM and 10 µM rosiglitazone in order. Differentiated fibroblasts from each passage were stained with LipidTOX to assess the level of differentiation. Stained neutral lipids are presented in green (Figure 5B). Particularly at passages 7 and 9, fewer lipid droplets were accumulated. The expression of LipidTOX was notably higher at passage 1, whereas its expression was observed at a lower rate at passage 9. The lipid droplet size of adipocytes from three passages were measured and compared (Figure 6A). Lipid droplet was also significantly the highest at passage 1 (*p* < 0.05). These findings were consistent with those obtained through flow cytometry analysis (Figure 6B,C). As passages increased, there was a significant decrease in neutral lipid accumulation within adipocytes (*p* < 0.05).

## 4. Discussion

In the process of isolating satellite cells for cultured meat production, fibroblasts are the main contaminating cell type. Fibroblasts could adhere to the flask and proliferate more quickly than satellite cells. Pre-plating is one of the processes used to isolate cells based on this difference in the adhesion time of cells [21]. In this study, fibroblasts were isolated and cultured by removing slowly adhering cells with pre-plating. Cells attached to the flask during pre-plating were utilized as CEFs. Primary CEFs and CSCs isolated by 2 h of pre-plating were morphologically indistinguishable until the second day of culture. However, fibroblasts were observed to adhere by spreading on the bottom surface of the flask. After the induction of differentiation, distinct morphologies between these cells became evident. Especially, CSCs formed myotube-shaped long tunnels through differentiation.

Also, a totally different doubling time was observed after 2 h of pre-plating for CSCs, a major muscle cell. In other words, this indicates that CEFs were isolated well from heterogeneous cells containing CSCs.

To identify isolated fibroblasts, CEFs and slowly adhering cells were immunofluorescence-stained with pax7. Pax7 is a distinctive transcription factor prevalent in satellite cells [31,32]. It is expressed during their quiescent phase and then gradually decreases [33]. CEFs attached to the flask within 2 h exhibited significantly lower expression rate of Pax7 compared to CSCs, which took longer to attach (*p* < 0.05). A 40-fold difference in pax7 expression rates indicated that CEFs could be effectively isolated from muscle tissue which CSCs are abundant through pre-coating. The results of staining cells with fibroblast markers showed that over 90% of rapidly adhering cells within 2 h were fibroblasts. CD29, also known as integrin beta1, and vimentin are cell proteins that can serve as markers for chicken fibroblasts [10]. This demonstrates that CEFs can adhere to flasks coated with collagen within 2 h.

Fibroblasts have an advantage over satellite cells primarily utilized in cultured meat production by providing a more stable cell supply due to their rapid proliferation capacity. With the aim of leveraging benefits of fibroblasts for cultured meat production, the proliferative potential of CEFs was analyzed. Doubling time, which is required for the number of cultured cells to double, signifies how many passages isolated primary fibroblasts can sustain their proliferation capacity and how much time fibroblasts take to double in vitro [34]. Initially, due to the impairment of various enzymes, attachment proteins, and substances involved in cell interaction, the doubling time was significantly high at passage 0 [35]. However, as cultivation commenced, the cells stabilized their proliferation and showed consistent growth capability. Based on the analysis of population doubling time and CCK absorbance results, it was observed that cell growth began to decelerate, and viability started to decline with continued cultivation from at least passage 11 onward. Furthermore, this aligned with the results of EdU assays, which did not show significant differences in proliferation capacity up to passage 9. In the EdU analysis under the same conditions, fibroblasts from passage 11 showed scarce cells expressing EdU. These comprehensive results of proliferation analysis indicate that CEFs proliferated very slowly or ceased proliferation from passage 11 onward. In other words, CEFs can consistently survive and proliferate up to passage 9 in vitro. This was supposed to be the loss of stemness due to senescence in primary cells [36]. The loss of stemness in primary fibroblasts can significantly impact the yield and cost of cultured meat, which are its weaknesses compared to conventional meat [37]. The decline in cellular productivity is directly related to the cost of mass production, which is a critical issue in the cultured meat industry. Our comprehensive proliferation analysis demonstrated the stable proliferation of CEFs and a cumulative cell number of 46 × 10^12^ up to passage 9. The observation that primary CEFs achieved stable proliferation up to the ninth passage with only 20% FBS, without additional aid of other growth factors, indicates a high level of productivity from a food production perspective. These outcomes suggest the potential application of CEFs to be utilized as cultured meat components.

Oleic acid is a fatty acid capable of inducing the differentiation of chicken preadipocytes into adipocytes [25]. It has been reported that oleic acid can elevate lipid droplet accumulation and regulate the activity of glycerol 3-phosphate dehydrogenase (GPDH) involved in lipid synthesis [38,39]. It can also upregulate fatty acid-binding protein (FABP), a lipid mediator gene. Particularly, oleic acid or oleate is predominantly utilized in the adipogenic differentiation of mouse and chicken cells via trans-differentiation [25]. During the initial three days of differentiation, 400 µM oleic acid was added to DM. The addition of oleic acid induced morphological changes in fibroblasts, transitioning them into preadipocytes and starting lipid droplet formation. The large size of the lipid droplets is specific to the morphologies of mature adipocytes [40]. However, as fibroblasts maintained their fusiform shape and failed to accumulate sufficient lipid droplets, the addition of only oleic acid was insufficient to induce complete trans-differentiation into mature adipocytes. There is also a previous study in which the addition of oleic acid alone did not fully induce adipogenic trans-differentiation in chicken fibroblasts [10]. There is a report that chicken preadipocytes can be induced to change into mature adipocytes with the addition of rosiglitazone to DM [41]. Rosiglitazone, a member of the thiazolidinedione family, functions as a peroxisome proliferator-activated receptor-gamma (PPARγ)-specific agonist capable of robustly inducing differentiation into adipocytes [15,23]. PPARγ acts as a master regulator of adipogenesis, playing a decisive role in adipocyte differentiation [42,43]. It has been reported that culturing chicken fibroblasts in differentiation media supplemented with rosiglitazone can significantly increase the activity of factors related to peroxisome proliferator response [38,41]. Hence, rosiglitazone was supplemented in the later stages of differentiation to facilitate the differentiation of adipocytes into mature adipocytes in conjunction with oleate activity.

By the seventh day of differentiation, cells transformed from fusiform to round-shaped mature adipocytes laden with lipid droplets. Adipocytes from passage 1 exhibited relatively rounder and larger morphology compared to those from passage 9, attributed to an increase in lipid droplet size. Adipocytes from passage 9 exhibited a spindle shape like fibroblasts [44]. This is attributed to the fact that fibroblasts at higher passages possess lower adipogenic differentiation capability compared to fibroblasts at lower passages, resulting in slower differentiation. Particularly when DM was applied to the cells, cell death occurred frequently at higher passages like passages 9 and 11. This is believed to stem from the adipogenic induction substances and the DMSO used as their solvent in DM. It was supposed that cells at higher passages exhibited heightened sensitivity to these differentiation factors and have lower viability. The lower cells viability due to differentiation-inducing substances in the culture medium, along with the progressed passage, is likely a natural consequence of the senescence of primary cells [45]. These results imply that decreased lipid droplet accumulation due to cell senescence at high passages can prevent complete differentiation. Similarly, decreases in adipogenesis were reported in a previous research study about human adipocytes due to activin A secreted from senescent cells [46]. In this manner, it is essential to further investigate the causes of reduced adipogenesis due to cellular senescence in chickens. This will enhance our understanding of the mechanisms involved and potentially identify targets for mitigating the impact of senescence on adipose tissue function. The results of the study also imply that growth factor supplementation or the use of specific cell lines needs to be considered to sustain the viability of primary fibroblasts during adipogenesis for cultured meat production.

## 5. Conclusions

This study has shown that chicken embryo fibroblasts (CEFs) can be isolated by 2 h of pre-plating. While primary CEFs can survive during 15 passages, proliferation capacity was decreased at passage 11. Maintaining the proliferation and stemness of CEF stable was possible up to passage 9 due to the senescence of CEFs. Primary fibroblasts which can maintain their cellular characteristics and differentiation potential even at high passages could serve as a valuable cell source for the cultured meat industry. Also, cell senescence prevented the complete adipogenic trans-differentiation of CEFs. This means that as passages progressed, adipogenesis was decreased by the senescence of primary cells. The decrease in the adipogenic trans-differentiation rate of primary CEFs as the passage of cells increases can be a critical challenge to the cost and production efficiency of the cultured meat industry. To prevent the decline of cultured fat yield due to primary cell senescence, growth factor supplementation or the usage of cell lines must be considered for cultured meat production. Therefore, systematic cell management strategies and approaches to maintain cellular characteristics over the long term are essential to maximize the stability and efficiency of cultured fat production for the cultured meat industry.

## Figures and Tables

**Figure 1 cells-13-01734-f001:**
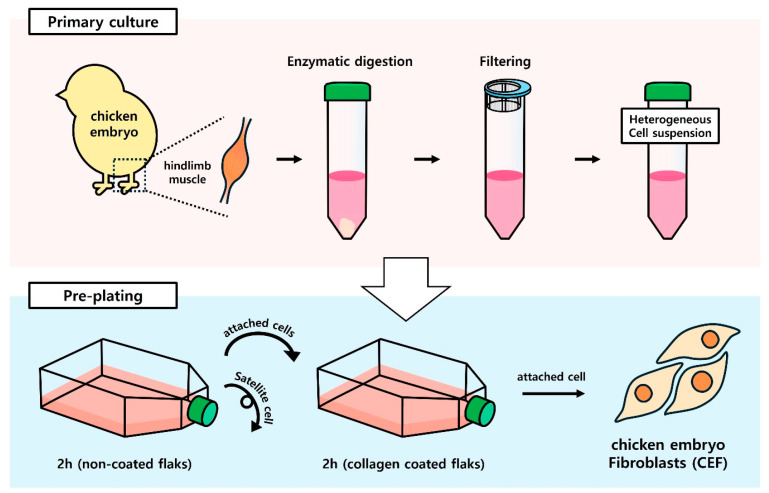
Schematic illustration of CEF isolation from chicken embryo hindlimb muscles by pre-plating.

**Figure 2 cells-13-01734-f002:**
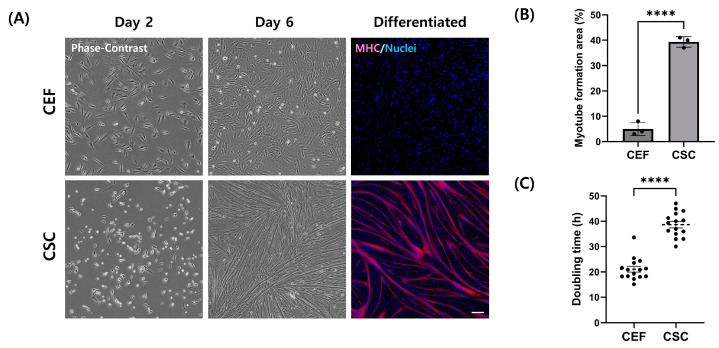
Analysis of the characteristic differences between CEFs and CSCs from passage 0. (**A**) Representative phase-contrast and staining images of CEFs and CSCs. Cells were immunofluorescence-stained with MHC (red) and Hoechst 33342 (blue). Scale bar = 100 μm. (**B**) Difference in MFA between CEFs and CSCs. Mean MFA was calculated from five randomly selected images. (n = 3) (**C**) Doubling time of primary CEFs and CSCs. (n = 15) Data are presented as means plus standard deviation from three independent experiments. **** indicates statistically significant difference at *p* < 0.0001.

**Figure 3 cells-13-01734-f003:**
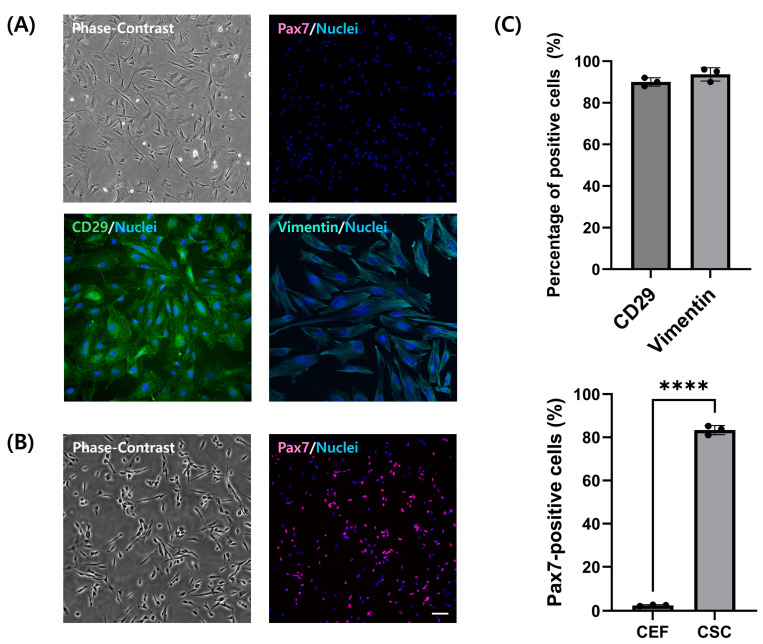
Identification of CEFs using immunofluorescence staining. (**A**) Representative images of CEFs stained with pax7 (yellow), CD29 (red), and vimentin (cyan). (**B**) Phase-contrast and staining image of CSCs. CSCs were stained with pax7 (red). (**C**) Percentage of cells expressing CD29, vimentin, and pax7. Scale bar = 100 μm. (n = 3) Data are presented as means plus standard deviation from three independent experiments. **** indicates a statistically significant difference at *p* < 0.0001.

**Figure 4 cells-13-01734-f004:**
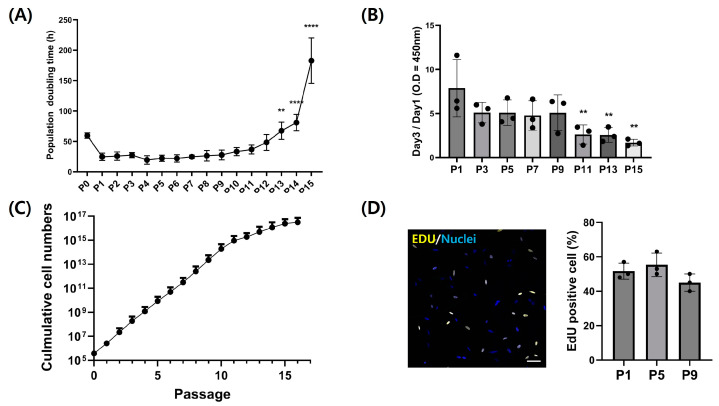
Proliferation capacity of CEFs during long-term culture. (**A**) Population doubling time of CEFs cultured up to passage 15. (**B**) CCK-8 absorbance graph of CEFs of 3-day culture by passage. (n = 3) (**C**) Growth rate of CEFs calculated using cumulative cell numbers. Data are presented as mean ± standard deviation from three independent experiments. (**D**) Representative image and percentage of EdU-positive cells. Cells were immuno-stained with EdU (yellow) and Hoechst 33342 (blue). Scale bar = 100 μm. (n = 3) Data are presented as means plus standard deviation from three independent experiments. ** and **** indicate significant differences at *p* < 0.01 and *p* < 0.0001, respectively.

**Figure 5 cells-13-01734-f005:**
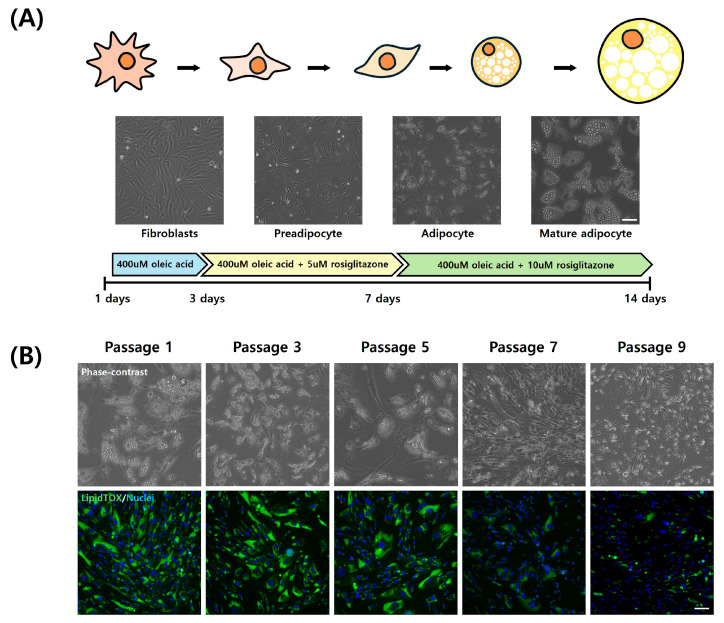
Trans-differentiation of chicken fibroblasts into adipocytes during long-term culture. (**A**) A schematic diagram of the trans-differentiation of CEFs into mature adipocytes by DM change. (**B**) Representative images of trans-differentiated CEFs by passage. Cells were stained with lipidTOX (green) and Hoechst 33342 (blue). Scale bar = 50 μm.

**Figure 6 cells-13-01734-f006:**
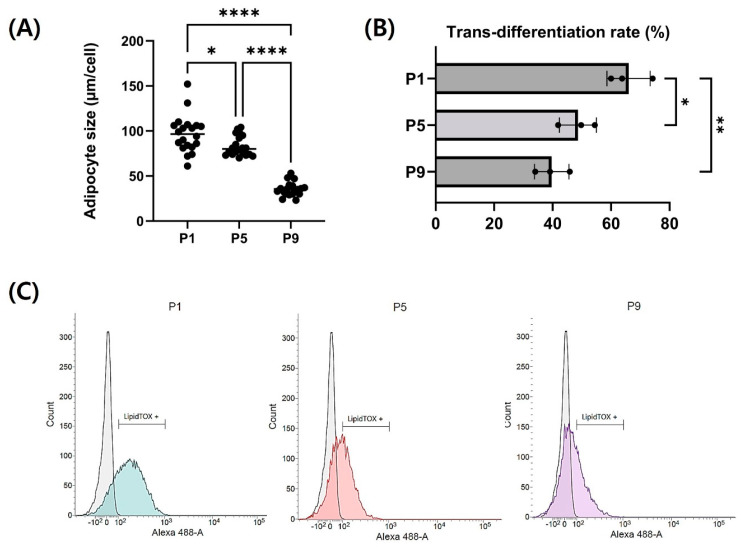
Analysis of adipogenic trans-differentiation level of CEFs at P1, P5, and P9. (**A**) Difference in adipocyte size by passage. (n = 20) (**B**,**C**) Trans-differentiation level by passage with flow cytometry during long-term culture. The zone within the gate-bar means LipidTOX-positive cells. (n = 3) Data are presented as means plus standard deviation from three independent experiments. *,** and **** indicate significant differences at *p* < 0.05, *p* < 0.01 and *p* < 0.0001, respectively.

## Data Availability

Data will be made available on request.

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
