# Peer review of "Chicken Embryo Fibroblast Viability and Trans-Differentiation Potential for Cultured Meat Production Across Passages"

_cells, 2024, doi:10.3390/cells13201734_

Round 1
Reviewer 1 Report
Comments and Suggestions for Authors
The article describes a study that analyzes the viability of chicken embryo fibroblasts and the efficiency of their transdifferentiation into fat cells for cultured meat production. It was demonstrated that chicken embryo fibroblasts can be stably isolated and undergo adipogenic transdifferentiation, indicating their potential use in artificial meat production.
The results are presented in an interesting form, however, the Authors should pay attention to minor errors:
- The quality of the drawings needs to be improved, many of them are illegible, especially the scale in figure 4a
- figure 3 is in the middle of the description, it should be properly cited and placed in the right place. Currently, the description which is separated by the drawing is not very readable
- the conclusions should be expanded
Author Response
Manuscript Title:
Chicken Embryo Fibroblast Viability and Trans-differentiation Potential for Cultured Meat Production Across Passages
#Reviewer 1
We sincerely appreciate the thoughtful review you have provided. Thanks to your valuable feedback, the quality of our manuscript has significantly improved. Below, we have outlined our responses to your comments. We have incorporated most of your suggestions and made revisions throughout the manuscript accordingly. Thank you once again for your insightful guidance.
- The quality of the drawings needs to be improved, many of them are illegible, especially the scale in figure 4a
: As you mentioned, we have adjusted the layout and size of Figure 4 to enhance its overall visibility, particularly improving the legibility of the scale in Figure 4a.
- figure 3 is in the middle of the description, it should be properly cited and placed in the right place. Currently, the description which is separated by the drawing is not very readable
: Throughout the revision process, we have carefully adjusted the placement of figures and rearranged the paragraphs to enhance the overall structure of the manuscript. If you have any specific suggestions for a better placement, we would be happy to change the manuscript.
- the conclusions should be expanded
: Thank you for your valuable suggestion regarding the expansion of the conclusion. We have supplemented the missing parts of the original conclusion and included recommendations for collaborative efforts between academia and industry based on the findings of this study.
Reviewer 2 Report
Comments and Suggestions for Authors
1. Abstract should be more precisely with better conclusion not only results.
2. At the begging of material and methods should be a short description of experiments which is in results section and every step of materials and methods should be expanded about aim of presented step.
3. Text should be aligned to margins
4. Discussion should be in separate section
Author Response
Manuscript Title:
Chicken Embryo Fibroblast Viability and Trans-differentiation Potential for Cultured Meat Production Across Passages
#Reviewer 2
We sincerely appreciate the thoughtful review you have provided. Thanks to your valuable feedback, the quality of our manuscript has significantly improved. Below, we have outlined our responses to your comments. We have incorporated most of your suggestions and made revisions throughout the manuscript accordingly. Thank you once again for your insightful guidance.
- Abstract should be more precisely with better conclusion not only results.
: Thank you for your insightful suggestion. Considering the word limit for the abstract, we have made slight adjustments to the existing content and incorporated the conclusions accordingly.
- At the begging of material and methods should be a short description of experiments which is in results section and every step of materials and methods should be expanded about aim of presented step.
: We appreciate your valuable advice. A brief description of the experiments, as suggested, has been provided throughout the Materials and Methods section. Additionally, we have revised the section titles to reflect the purpose of each experimental step more clearly.
- Text should be aligned to margins
: Following your recommendation, we have adjusted the alignment and margins across the entire manuscript. Thank you for your helpful feedback.
- Discussion should be in separate section
: In response to your suggestion, we have separated the results and discussion into distinct sections.